# Characterization of PS/PP/HDPE/LDPE Polymer Blend Obtained from Plastic Waste Collected on Beaches in Ilhéus-Bahia, Brazil

**DOI:** 10.3390/polym15204155

**Published:** 2023-10-19

**Authors:** Tauane Winny Silva de Jesus, Daniel Pasquini, Tatiane Benvenuti

**Affiliations:** 1Post Graduation Program on Science, Innovation and Modelling Materials, Department of Exact and Technological Sciences Rodovia Jorge Amado, Santa Cruz State University, Ilhéus 45662-200, Brazil; tbenvenuti@uesc.br; 2Institute of Chemistry, Federal University of Uberlândia, Av. João Naves de Ávila, 2121, Bloco 1D, Campus Santa Mônica, Uberlandia 38400-902, Brazil; daniel.pasquini@ufu.br

**Keywords:** marine pollution, polymers, recycling, properties, morphology

## Abstract

A large volume of polymeric waste is generated in cities, and some of this reaches the sea and beaches. This waste stays for hundreds of years, damaging marine environments and organisms. To minimize the effects of pollution, collection and recycling allow a return to the production chain. This research aims to produce and evaluate a polymeric mixture obtained via processing plastic waste collected on the beaches of the city of Ilhéus-Bahia. Subsequently, the mixture is converted into a granulated form for application as fine aggregate in the production of cementitious matrices. A polymer blend of polystyrene, polypropylene, and high- and low-density polyethylene was obtained and evaluated by thermal, morphological, and mechanical tests in three processing stages. The degradation temperatures were close for the three processing stages and the level of processing influenced the mechanical strength. As for elastic modulus and deformation, there was no significant difference in using the mixture processed once or twice. The results showed that the reuse of the waste is applicable, the mixture presented a compact, reasonably homogeneous material with different morphology. Therefore, this work finds importance in the possibility of promoting waste recycling and adding value to a material that would become waste, thus showing its potential for application in the construction industry as an addition to cementitious mixtures and leading to savings in inputs.

## 1. Introduction

Plastic materials are among the most consumed today, due to their great molding versatility, lightness, and resistance to impact and bacteria. Disposable products and food, electronic packaging, and domestic and industrial equipment are produced from this input [1,2]. In Brazil, plastic production reached 7.1 million tons in 2021, generating a turnover of BRL 127.5 billion [3].

However, the big problem with this material is improper disposal after consumption. As it is not biodegradable, it takes hundreds of years to decompose, causing damage to the environment. From disposal in dumps or reaching rivers and beaches, such waste can arrive at the oceans, be ingested by marine animals, and, until their degradation, form islands of tailings on the high seas, turning into microplastics. The garbage present on the beaches is a problem that transcends borders, generates environmental impact, and puts risk to marine life and water quality.

Research has been developed to measure the level of pollution by solid waste present on the beaches of Brazil [4,5,6,7,8]. Their analyses corroborate the claim that plastic debris is the majority (average of 75%) in the marine environment. Many of them, with an elevated level of degradation, have already become microplastics. Additionally, the main sources of waste are beach users, domestic garbage, and urban drainage.

Due to the harm, there is an urgent need to find mechanisms to minimize the effects of pollution, recycling, reusing, or incorporating plastic waste into new materials. The reuse method of these materials must consider the characteristics, the contamination level, and the type of mixed material. Among the reuse methods most applied is recycling (chemical, mechanical, and/or energy recovery). Its principle is based on the transformation of waste into new products from processing techniques, aiming at the alteration of its biological, physical, or physicochemical properties [2,9,10].

In this work, the method chosen to conduct the reuse of plastic waste was mechanical recycling (secondary). It is the most common method for post-consumer waste. Often these materials have an elevated level of contamination and degradation and are mixed, demanding pre-treatment, classification, and decontamination steps. Through a mechanical process (extrusion, injection, blowing, among others) or a combination of them, the waste material is converted into a new product. In this case, the performance properties are reduced on the original product [11]. Usually, a polymeric blend is obtained from this process.

Polymer blends are formed via the combination (physical or mechanical) of two or more different polymers to obtain a new material with improved properties or to promote the recycling of post-consumer polymer blends, reducing production costs. They can be prepared by dissolving the polymers in the same solvent or by melting the mixture at a working temperature that does not cause degradation [11,12,13]. A common and low-cost method to obtain blends is by melting compounds in extruders [14].

The extrusion process consists of converting a raw material, normally in powder or grains, into a molten mass (finished or partially finished product), through its softening and compression by an orifice. This technique is widely used in obtaining tubes, rods, films, polymer blends, primary products, and the recovery of wasted materials [15,16].

Extrusion is applied for processing blended polymers. The heterogeneity of the sample is related to the degradation process that the material undergoes during the pre-treatment steps, such as milling and drying, and results in shear, thermo-oxidative, and hydrolysis degradation. Degradation affects properties such as appearance, chemical resistance, and mechanical properties of polymer blends [16]. Therefore, for the processing of polymeric materials via extrusion, thermal properties such as degradation temperature, melting temperature, and glass transition temperature must be taken into thermal properties such as degradation temperature, melting temperature, and glass transition temperature must be taken into consideration when finding the temperature profile for heating zones.

For polymeric mixtures, two properties are essential to characterize the resulting material and definition of your behavior: the level of interaction between the blend constituents and the morphology (do-main/drop, co-continuous, among others), related to the viscosity degree. The morphology is determined via the processing conditions, mixture composition, rheology, and interfacial tension. It also can be characterized via microscopy techniques, such as scanning electron microscopy (SEM). Regarding the level of interaction between the components of the mixture, blends can be miscible or immiscible [17,18].

Another variable that must be considered in the production of polymeric blends is the polymer degradation level, which is a result of the type of material processing. Degradation affects properties such as appearance, chemical resistance, and mechanical properties of polymeric mixtures [16,19]. Thermal analyses such as thermogravimetry (TGA) help to identify the material’s decomposition temperature. During extrusion, for example, the material is subjected to elevated temperatures and shear rates, which leads to the combined effect of thermal, mechanical, and oxidative degradation. Therefore, the properties of the polymeric material found in waste, the recycling techniques used to conduct their reuse, and the level of degradation suffered in the process, are important parameters to characterize and understand the microstructure of the obtained material.

From what was presented, this work aims to produce and evaluate a polymeric blend obtained through the processing of mixed plastic waste, which was collected on beaches in the city of Ilhéus-Bahia, and to promote its processing as a fine aggregate in the production of the cement matrix.

## 2. Materials and Methods

Figure 1 presents the developed method to obtain a polymeric blend from post-consumer plastic waste, its characterization through thermogravimetric tests, tensile strength, morphology, and its processing as recycled aggregate.

### 2.1. Waste Collection and Preparation

The collection of polymeric wastes used to obtain the blend of polymeric was conducted in the city of Ilhéus-BA, Brazil. The Praia do Sul Beach (−14.819681857699996 S, −39.024820241836224 W), Praia da Avenida Beach (−14.801273667201308 S, −39.0298465940346 W), and Malhado Beach (−14.784590625305476 S, −39.037764506186576 W) were the collection points chosen, due to the greater flow of visitors/users. The collection period occurred from December 2020 to May 2021.

The collection was conducted on the sand strip between the waterline and the upper limit of the beach environment. The beach cleanings covered a longitudinal stretch of about 1.0 km. All waste found on the beaches was collected manually. The materials of interest for this research (polymeric residuals) were separated into bags for transport. In Figure 2, the collected waste (Figure 2A) is prepared for transport (Figure 2B), and the washing step preview (Figure 2C,D).

The collected materials were grouped and submitted to the prior washing process, as it was contaminated with organic residues, sand, and salts due to contact with seawater. A polyethylene tank with a capacity of 1000 L and potable water was used in this step. The waste was added to the reservoir and remained immersed for about 3 h. After this period, the materials were removed from the tank and subjected to drying step for about 12 h.

The identification and classification of each waste were conducted visually from the identification symbols presented by the standard NBR 13230—Indicative Symbology of Recyclability and Material Identification Plastics [20]. The waste was separated according to the type of plastic and weighed to set up the proportions of each polymer. The digital scale used was from ELGIN brand, model 1502, with weighing capacity of 15 kg.

The sample components that were heavily contaminated or with an elevated level of deterioration, with visual identification not possible, were classified and put together in a separate item, and these materials were then discarded from the set.

### 2.2. Milling and Homogenization

The dry material was initially fragmented using scissors to reduce its size to ease the next stage of recycling. Subsequently, the milling was conducted in a knife mill (RONE line N, model N150). The material to be milled was placed in the feed nozzle located at the top of the equipment, and, to assist milling, a stick was used to press the material into the equipment since, due to the size of the waste and low weight, the material accumulated inside the feed channel.

Milling occurred by type of material, clean and dry, and it was carried out in several crushing stages. The material re-feeds the mill 3 to 4 times, on average, to obtain particles of uniform dimension. The entire ground sample was homogenized manually.

### 2.3. Washing and Drying

To ensure efficient cleaning, the ground sample was subjected to the washing and drying process. After homogenization, it was placed in a washing tank with a capacity of 50 L, and water was added to the container until the sample was completely submerged. The mixture of milled polymers was filtered through a polypropylene mesh with an opening of 3 mm. The environmental conditions were the same as in the previous drying. However, after 12 h, the material was submitted to outdoor exposure to the sun for another 5 h.

### 2.4. Preparation of Blends

A sample of the milled material was subjected to processing in the extruder, a small size equipment of model EFP01 single screw, with temperature control in the two heating zones (nozzle and center). The temperature for each zone was determined according to the intermediate melting temperatures of the polymers present in the sample (PP, PS, HDPE, and LDPE), and 190 °C was set up in the center and 200 °C in the nozzle. Although the melting temperature of polystyrene (240 °C) is higher than that of the other polymers, this was not accounted for because the number of PS in the sample was minimal (1% of total mass). In addition, as its glass transition temperature is lower than the melting temperature, heating to the set temperature would cause partial melting of the material.

After processing, a trough filled with water received the extruded material and cooled it down to room temperature (~25 °C) to ensure dimensional stability. The product of extrusion processing is what is called a polymer blend. To evaluate the effect of processing at different extrusion stages on the final properties of the blends, a part of the samples obtained from the first (1st) extrusion was also subjected to further processing (2nd extrusion) under the same condition.

### 2.5. Characterization of Blends Polymeric

The produced polymer blends were characterized via thermogravimetric analysis, mechanical strength, and scanning electron microscopy to verify the composition and the effect of the recycling processes on the polymer properties after 1st extrusion and reprocessing (2nd extrusion).

These analyses supply important parameters to evaluate the level of degradation of the samples under the action of temperature, mechanical strength, and morphological analysis on the rupture surface, and thus determine the best processing cycle for introduction into the concrete.

#### 2.5.1. Thermogravimetric Analysis (TGA)

The polymeric mixtures were evaluated and characterized via thermogravimetric analysis. The equipment used was the Shimadzu DTG-60H (Barueri, Brazil), in the presence of an oxidizing atmosphere (O_2_) with a flow of 30 mL min^−1^. The sample, with a mass between 5–7 mg, was placed in an aluminum sample holder at a heating rate of 10 °C min^−1^, in the range of 25 to 600 °C. For this test, the ground, extruded sample (1st extrusion), and reprocessed (2nd extrusion) were analyzed.

To ensure the same heating conditions and homogeneity between the material samples (before processing in the extruder, after first processing, and after second processing in the extruder), each fraction was ground in a knife mill to reduce particle size.

#### 2.5.2. Tensile Test

The tensile strength test was carried out following the standard ASTM D882 [21]. To obtain the specimens, the extruded material was subjected to hot compression using a thermal press model MPH-10 from MARCON hydraulic line.

Using a 5 kN load cell and a test speed of 25 mm·min^−1^ (elongation rate) in the Instron 5982 universal testing machine. After conditioning for 24 h at 25 °C, six specimens (geometry—1 cm × 11 cm) of each extrusion were evaluated at 25 °C. Under these conditions, the following were obtained: tensile strength, elastic modulus, and deformation.

We used an Instron tensiometer model M20-52630 with length equal to 132 mm and maximum displacement of 25 mm. To obtain the specimens, the extruded material was submitted to hot compression using a thermopress, model MPH-10, from MARCON’s hydraulic line. For the molding, two Teflon plates with dimensions 175 × 170 × 3 mm^3^ were used as molds.

The extruded material was placed between the plates and coupled to the equipment. O process was conducted by heating the apparatus up to a temperature of 140 °C and kept at this molding temperature for 10 min. This temperature (below the extrusion temperature) was used due to the limitation of the press that allowed heating at a temperature of approximately 140 °C.

To verify the experimental results obtained in the test of tensile strength, the Chauvenet criterion was used. The influence of reprocessing the polymeric blends was evaluated from the analysis of variance (ANOVA). In this research, ANOVA was applied to choose the processing (1 or 2 extrusions) that would be used for polymeric blend production for incorporation into concrete. The software used was STATISTICA 7.0, and the confidence level adopted was 95%.

#### 2.5.3. Scanning Electronic Microscopy (SEM)

The polymeric blend obtained from the 1st and 2nd extrusions was manually broken, and the fracture surface was analyzed via scanning electron microscopy (SEM). The SEM analysis was performed on a ZEISS device, model EVO MA10. The goal of this analysis is to evaluate the sample surface, as well as the effect of reprocessing on the morphology of the polymer blends.

In a vacuum-metalizing chamber, the samples were coated by a layer of gold (e = 40 nm), and the magnifying chosen were 50, 200, 1000, and 5000 times. As a comparison, the morphological analysis was performed on two types of samples for each extrusion. In the first sample is the extrudate processed with the extruder nozzle, and in the second, this nozzle was removed from the extruder outlet, obtaining a larger diameter spaghetti. The particle size of the dispersed phase was measured from the images obtained by SEM using the ImageJ software.

#### 2.5.4. Production of Fine Aggregate

After processing extrusion, the recycled material was subjected to milling for transformation into fine aggregate. The sample passed three times in a disintegrator and particle grinder, model DPM-JÚNIOR, with a motor power of 2.0–3.0 HP. The extruded material was introduced into the feeding channel, located in the upper part of the equipment, and submitted to the processing. In the milling process, the internal sieve size varied (10.0 mm, 8.0 mm, and 5.0 mm). This reprocessing was necessary to obtain a uniform and sand-like particle size used in the production of cementitious materials. After grinding, the material went through sieving on a sieve with an aperture of 4.75 mm.

## 3. Results and Discussion

This topic will address the results found for the collection, treatment, and characterization of polymeric waste from beaches. Besides the characterization tests of the recycled polymeric material, namely, thermogravimetry, tensile strength, and morphology.

### 3.1. Polymeric Waste Collection and Characterization

Despite the restrictive measures due to the advances of the pandemic in the city of Ilhéus-BA, there was a concentration of solid waste present on the beaches, mainly after periods of rain, which indicates the origin of these materials: urban environment, brought by rainwater drainage, added to irregular sewage and domestic disposal systems—in addition to the beach user waste.

Meal packaging in EPS (expanded polystyrene) and cleaning products, such as alcohol and bleach, were recurring items in the collections provided by the pandemic. With the restrictive decree’s circulation and curfews, people stayed longer at home, promoting an increase in the use of delivery services. The concern with the cleanliness of environments, surfaces, and hands to reduce exposure to the virus promoted large-scale consumption of sanitizing products, whose packaging was easily found on beaches.

After the collection, transport, and storage of plastic waste, the prewashing and drying steps took place. It was possible to conduct the preventive cleaning of inputs and total drying in a period of 12 h. The polymeric materials were classified and weighed separately according to their type. PET materials were not considered for the research because they are recycled more easily and sold by a cooperative of collectors in the city of Ilhéus (COOLIMPA) and due to their high melting temperature in relation to the other selected polymers. In Table 1, there are the types of collected plastics, waste characteristics, and their respective quantities.

The sample selected for this study is composed of 54% PP, 39% HDPE, 6% LDPE, and 1% PS (exclusively from plastic cups). The most common waste, concerning the number of items collected, were the pots and lids of margarine/vegetable cream-type foods and disposable cups. The materials classified as OTHERS were discarded from the sample as these are unidentified fragments or residuals with unrecognized composition, therefore, limiting the process of material recycling.

### 3.2. Milling, Washing, and Drying Polymeric Waste

Despite the wide variety of forms and types of polymers, the milling of the material was satisfactory, and the material with a granulometry of up to 5 mm was obtained. Washing and drying the crushed plastic waste sample allowed us to obtain a sanitized and dry material.

### 3.3. Characterization of the Processed Polymeric Material

After conducting the plastic waste purification steps, the material was ground and extruded. In this topic, the results obtained in the characterization tests of the recycled polymeric material, namely, thermogravimetry, tensile strength, and morphology will be presented.

#### 3.3.1. Extrusion Processing

Figure 3 shows the ground material after the first and second extrusions. It was possible to carry out the processing of the material with the formation of a completely homogeneous molten mass with a rough surface. The continuity of the filament was verified without ruptures or breaks, which proves the mixture’s stability. The swelling of the extruded material at the exit of the matrix was also verified due to the elasticity of the polymer. It was visually saw the presence of pores in the cooled material was reduced in the second process; however, the material still showed some porosity.

Mechanical recycling through the extrusion of blended polymers proved to be useful for plastic waste processing. It was obtained from a uniformly colored material (greenish gray), agglutinated, interacting between the components of the mixture and with a minimum number of unfused particles of polymeric materials.

#### 3.3.2. Thermogravimetric Analysis

Regarding the results obtained for the present study, Figure 4 presents the thermogravimetry (TG) and derivative thermogravimetry (DTG) curves for the polymer blend (PS/PP/HDPE/LDPE) ground (initial polymer) after extrusion and reprocessing.

For the material obtained after milling, the degradation process started at around 239 °C and ended at around 510 °C, presenting a mass loss of about 96%. For the blend obtained in the first extrusion, the degradation started at around 247 °C and concluded at around 500 °C, presenting a mass loss of about 97%. The reprocessed polymer blend (2nd extrusion) started its degradation at around 250 °C and ended at around 490 °C, showing about 98% of mass loss. The intensity of peaks in the DTG curve, where the highest degradation rates occur, rises progressively as the cycle increases.

The blend submitted to milling started its degradation process at a low temperature and finished at a higher temperature compared to the extruded materials, signaling a higher reaction speed and less stability. The result differs from the expected one, as this material was not submitted to any thermal process before the test, like the other samples, and therefore should be the most stable. However, it reflects the interference of the compaction, density, and shape of the original sample in the heat transfer and, so, the mixture stability. Thus, the specificity of the first sample makes the comparison with the others too much limited.

Table 2 shows parameters such as the initial temperature of degradation (T_i_), which indicates the beginning of volatile output and, therefore, the degradation process, the final degradation temperature (T_f_), which signals the conclusion of the mass variation process, the percentage of mass loss at each stage (%), the temperature of maximum degradation (T_max_), defined by the highest intensity peak in the DTG curve and the ΔT, which reflects the kinetics of the mass loss process. For T_i_ determination, 2% of the loss from the initial mass was used as a criterion, a procedure used by other authors [22].

It is observed that a high mass loss (96%, 97%, 98%) shows almost complete material degradation (conversion to volatile). As there is a mixture of polymers with distinct colors, the not degraded percentage (2% to 4%) possibly refers to fillers and inorganic pigments present in the printing inks of the labels and the polymer’s constitution. Up to 200 °C, there was a small loss of mass (<2%) attributable to the moisture present in the samples. As reprocessing (initial, first, and second extrusion), the DTG curves became narrower and with peaks of higher intensity, indicating a homogenous distribution of the molecular chain.

Regarding peak identification, the variability of mixed polymers and the interaction processes that may occur make the evaluation more complex. Therefore, this research will be limited to identifying the peaks for the major components within the degradation range of each polymer. The minority components (PS and LDPE) that together represent about 7% of the sample in a universe of approximately 7 mg have their degradation peak practically imperceptible in the thermograms. PP represents about 54% of the sample, and HDPE, about 39%, is the major component.

For the initial mixture (after milling), the first decomposition peak occurs at a temperature of 324 °C and concerns a decomposition of 24% of the sample. At the second peak of greater intensity, 20% of the sample decomposes at a temperature of 362 °C. The third peak occurs at 451 °C and refers to a decomposition of 38%. The first and second peaks possibly refer to PP degradation. As it is the ground material, several types of packaging may have produced different shapes for the grains, and the degradation may have occurred at close stages. The last peak refers to HDPE degradation.

The resulting mixture from processing in the extruder (1st extrusion) presents three stages of degradation: at temperatures of 307 °C, with a decomposition of 7% at 370 °C, with a decomposition of 25%, and at a temperature of 444 °C, with a decomposition of 42% by mass, and a shoulder at 507 °C. Thus, the second peak refers to the degradation of PP, and the third peak, to the degradation of HDPE.

The mixture referring to the reprocessing in the extruder (second extrusion) degrades at a temperature of 370 °C (35% decomposition) and more intensely at 398 °C (29% decomposition) and 445 °C (26% decomposition). The first peak possibly refers to the degradation of PP. The second, to the degradation of HDPE, and the third, can represent the degradation of LDPE or HDPE/LDPE interaction.

Work carried out with individual plastic waste samples showed that PP, PS, HDPE, LDPE, and PET start the degradation process at around 300 °C and complete the decomposition at 500 °C, degrading in only one stage—the presence of only one peak in the DTG curve [23]. As for virgin materials, the literature shows higher degradation temperatures, showing greater stability of these polymers. Based on the thermogravimetric analysis, in virgin samples, PP and HDPE showed the same degradation interval (between 400 °C and 500 °C), with the maximum degradation temperature for PP being 447 °C and for HDPE, 467 °C. The thermal degradation temperature of virgin PS would be between approximately 350–500 °C, with the maximum degradation at 425 °C. For LDPE, the thermal degradation would be between 360–550 °C, with the maximum degradation between 469–494 °C [24].

Previous studies [25] analyzing plastic waste from PP, PS, HDPE, and PET when mixed observed, in polymeric blends constituted by PP/PS/HDPE, that the degradation of PS is delayed in the initial phase of the reaction due to the presence of HDPE and PP. As PS degradation begins, the formation of free radicals catalyzes the reaction, and degradation begins quickly. The decomposition of PS helps to reduce the temperature of reaction degradation. In the DTG curve presented in Figure 4, there is the appearance of two peaks, the first of PS degradation and the second of the fusion between the PP and HDPE peaks, showing the interaction between the two polymers. The effects of thermomechanical processing (extruder/injector) on the properties of polymer blends of PP/LDPE (virgin) and pure polymers were evaluated [26]. Samples were reprocessed zero (0), one (1), three (3), and five (5) times. Thermograms showed that the LDPE starts to degrade around 399 °C, while the PP decomposition is near 370 °C, making the LDPE more thermally stable. All blends follow a multi-step degradation process. As the processing cycle increased, the PP content functioned as a reducer of thermal stability, and the blend’s degradation curves approached pure PP. Regarding the peaks, LDPE has a narrower and sharper peak (more homogeneous molecular chain distribution), and PP has a broader peak (heterogeneous combination of the polymer chain). For the blends, the reprocessing made the curves wider, and the appearance of secondary shoulders (generation of smaller molecules and other volatile products) was observed.

Thermal degradation of the main constituents of the waste of a scrap shredder was evaluated by thermogravimetric analysis [27]. Among the materials found were PP, HDPE, LDPE, PS, and polyamide polymers. For individual plastic waste, it was verified that at a heating rate of 5 °C·min^−1^, the maximum rate of thermal degradation occurs at a temperature of 393 °C for PS, 460 °C for HDPE/LDPE, whereas the maximum degradation occurs at 438 °C for PP. Results obtained in the research developed for the degradation curves are close to the results found by other authors [23,24,25,27] and follow the same order of decomposition for the identified polymers—PS-PP-HDPE-LDPE.

It is important to note that the use of virgin plastic samples or recycled/post-consumer can generate different degradation ranges, mainly to the degradation process related to recycling. Virgin materials present higher values of degradation temperature and, therefore, greater stability. The sample format, the heating rate applied in the tests, the origin of the waste, the pre-treatments and preparation, and the type of processing carried out on the material also influence the result. When the sample source is municipal solid waste, this variability tends to increase.

#### 3.3.3. Evaluation of the Mechanical Properties of the Polymer Blend

After thermo-pressing the material, a film of a regular thickness (<1 mm) was obtained and cut into rectangular strips to perform the tensile test. The resulting material was compacted with a surface finish without having visible macropores. However, when broken, it was possible to verify pores with the naked eye in addition to the flexibility when handling. Figure 5 shows the ruptured specimens after the tensile test. Therefore, the process of obtaining the specimens was efficient. The data in Table 3 presents the mean results and standard deviation obtained for the tensile strength test. They show that, with reprocessing, there was a reduction in tensile strength (11%), and an increase in the modulus of elasticity (19%) and deformation (6%).

About the average values obtained, the number of processing cycles changes the properties of the blend. That is, the degradation promoted by a new extrusion changes the characteristics of the material, thus having a reduction in the tensile strength (from the first to the second processing). Table 4 presents the statistical analysis for the level of significance of 5% and a confidence interval of 95%. The results of the statistical analysis showed that the first and second extrusions formed a homogeneous group (statistically equivalent) for the deformation and elastic modulus, different from the tensile strength, which differs from each other (statistically different). 

From the consulted literary collection, no works were found that analyzed the mechanical properties of polymeric blends with the same residues used. Ref. [26] evaluated the effects of thermomechanical processing (extruder) on the mechanical properties of polymer blends of PP/LDPE (virgin) and pure polymers. The samples were reprocessed zero, one, three, and five times, and it was observed that the increase in the number of cycles (1×) of PP processing resulted in a reduction of 19% and 16% in tensile strength and modulus of elasticity, respectively. In the case of LDPE, there was an increase in tensile strength of 3% and 5% for the module. Increases that can be attributed to chain fission and thermomechanical degradation. For the blends produced, the multiple reprocessing led to a progressive reduction in tensile strength and deformation; the greatest reduction was for 5× processing. As for the elastic modulus, there was a slight increase because of the decrease in the general crystallinity of the mixtures.

A comparative analysis of the mechanical properties of blends produced with PS/LDPE, PS/PP, and PS/PMMA obtained with different proportions of polymers in an extruder was performed [28]. The results showed that for the tensile strength and tension in the rupture, the PS/PMMA blend obtained the best results, while the percentage of deformation was higher in the PS/LDPE blend. For the addition of 20% of the polymers in the PS matrix, values for stress-at-break properties, elastic modulus, and percentage of deformation, respectively, were approximately 46.0 MPa, 3.0 GPa, 1.50% for PS/PMMA; 30 MPa, 1.70 GPa, 3.70% for PS/PP; 28 MPa, 1.40 GPa, 5.1% for PS/LDPE [28].

Polymeric LDPE/PP mixtures of virgin resins and post-industrial plastic waste were also evaluated for their mechanical properties. For the samples processed in a single-screw extruder, the results obtained for the virgin polymers were approximately 205 MPa for the modulus of elasticity, 8.4 MPa for stress at break, and 92% for deformation. The blend produced with the residues presented values of 248 MPa, 10.5 MPa, and 383%, respectively. It is observed, therefore, that the treated residues produced a more attractive material [29].

The LDPE/HDPE/PP/PVC polymer blend was produced by mixing in solution, using xylene and tetrahydrofuran as a solvent, polyethylene-co-glycidyl methacrylate (PE-co-GMA) as a compatibilizer, and different loads of clay and dry wood floor. The results showed that the tensile properties were improved with the use of a compatibilizer due to improved interfacial adhesion. For the pure polymer blend, the values found were 5.24 ± 1.13 MPa for tensile strength and 84.38 ± 18.19 MPa for the elastic modulus. The mixture with PE-co-GMA obtained values of 8.46 ± 1.21 MPa and 113.25 ± 16.19 MPa, respectively [30].

A comparative analysis can also be carried out about the values defined by the literature for virgin polymers and those found in this research for the first and second extrusion, as shown in Table 5. It is observed that the results obtained are higher than for the tensile strength of LDPE. The modulus of elasticity presents higher values for the PS/PP/HDPE/LDPE blend produced than for the pure polymers of HDPE, LDPE, and PP. For the deformation, the values found are within the expected for the PS.

There is a dispersion among the results found in academic research for tensile strength, elastic modulus, and deformation. Thus, it is not possible to define a behavior pattern for the use of recycled polymeric material. This can be explained by the way the material was used, the contamination level of the sample, its heterogeneity, the types of polymers involved, the processing, and the level of degradation of the material used. Therefore, the specificity of each sample influences the result.

The measured values and the statistical analysis support the thesis that, for tensile strength, there is a significant difference in the use of the blend processed once or twice, having the best result for the first extrusion. As for the other properties, processing had no influence.

#### 3.3.4. Morphological Evaluation of the Polymer Blend

From the analysis of the fracture surface via SEM, the micrographs presented in Figure 6 and Figure 7 resulted in the material processed by one and two extrusions. Figure 6 shows the results for the first extrusion processing in the absence and presence of the extruder exit nozzle. Figure 7 presents the results for the second extrusion, without a nozzle and with the extruder nozzle exit, respectively.

The first verified detail, from the micrographs of the polymeric blends of PS/PP/HDPE/LDPE, is the containing voids in the microstructure of the materials (items A and C of Figure 6 and Figure 7). These spaces with elongated geometry may result from the release of volatile components due to the degradation process of materials during thermomechanical processing.

The morphological classification of the blend, according to the theoretical model presented by the authors [12,14,17,18,32], is very efficient when there are two types of polymers composing the blend because it is possible to identify the phases from the amount of each material. In this research, four types of polymers were used to produce the blend. Therefore, it is not possible to categorically classify the morphology. All samples have a multiphase character and a reasonably homogeneous appearance. In the first processing (Figure 6), it is possible to see the morphology of dispersed particles (drops) in a continuous matrix (domain).

Although the phases have not been analyzed separately, one can expect the dispersed phase to be composed of LDPE and PS; polymers present in smaller amounts in the mixture, the continuous phase being formed by PP and HDPE, major constituents of the sample. The reprocessed mixture (Figure 7), without the nozzle, indicates the presence of two morphologies, particles dispersed in a continuous matrix and a co-continuous one. Apparently, the beaked sample has matrix-fiber morphology.

Despite the heterogeneity of the blend composition, there is a surface of regular, continuous, and rough appearance on which the dispersed phase is distributed over the entire matrix. The adhesion presented by granules to the matrix, verified in the magnifications of 5000× (items B and D of Figure 6 and Figure 7), for all analyzed samples, suggests the partial compatibility of the components involved. PS, due to its reduced amount (1.0%), may still be lodged in the interfaces of the phases, acting as a compatibility agent or adhering to one of the phases.

Figure 6 shows the results for the polymer blend submitted to the first extrusion cycle. It appears that the use or not use of the nozzle at the exit of the extruder can influence the morphology of the resulting material. An increase in the number of voids and deformation of their geometries was observed, in addition to a greater dispersion of the drop phase in the domain, for the nozzle extrusion.

The expansion of the void area in the microstructure of the nozzle sample compared to the sample without a nozzle can be explained in terms of the greater degradation suffered by the blend due to the longer retention time. As the amount of material is reduced at the exit of the extruder due to the smaller diameter, the extrudate stays longer inside the screw being heated and mixed, which favors the miscibility and fusion of the particles. Allied with this, there is an increase in internal pressure due to compression in the orifice, which favors the shear. Consequently, it may undergo further decomposition. It also observed the appearance of fibrils in the sample processed with the exit nozzle of the extruder for the second extrusion cycle (Figure 7D).

Regarding the particle size distribution of the dispersed phase in the matrix, it is observed that the drops are spherical, rod-shaped, and elongated. They well adhere to the matrix phase and in variable sizes. For the first extrusion, without the nozzle, the average particle diameter is 0.32 µm, and with the nozzle, it is 0.25 µm. As for the second extrusion, the diameter for processing without the nozzle is 0.31 µm and 0.28 µm with the nozzle. A reduction in particle size is observed with the exit diameter of the extruder. That is, for processing with the nozzle (smaller area), there are particles of smaller diameter.

#### 3.3.5. Production of Recycled Aggregate

Based on the found results, it was decided to transform the material obtained in the first extrusion, without using the extruder nozzle exit into fine aggregate, and as it presented better results (mechanical strength, degradation level, and morphology). Figure 8 shows the evolution in the refinement of the recycled material, which was subjected to processing in the disintegrator.

The filament obtained in the extrusion was ground twice to achieve the desired granulometry. Thus, this step was efficient because a material of granulometry smaller than 4.75 mm was obtained, like a fine aggregate of medium granulometry, a material that can be used as a partial substitute for sand in the production of concrete and mortar.

## 4. Conclusions

This work aimed to carry out the collection of plastic waste on the beach of Ilhéus-BA and perform the characterization of the benefited material and its processing for use as fine aggregate. Considering the obtained results, the following conclusions could be reached:The washing, drying, and milling steps proved to be efficient. The most recurrent residues in the collections were the pots and lids of margarine/cream foods, disposable cups, EPS packages for meals, and cleaning products such as alcohol and bleach (provided by the pandemic);The sample collected, separated for analysis, was composed of 54% PP, 39% HDPE, 6% LDPE, and 1% PS;Through extrusion processing, a polymeric blend of PS/PP/HDPE/LDPE was obtained, with homogeneous characteristics, rough surface, uniform in color, and having voids;Regarding the morphology of the polymer blend, all samples presented a multiphase character with a regular-looking surface, continuous and rough;The results for elastic modulus, deformation, and resistance to tensile strength at rupture showed that the extruded material only presented better performance to be used as aggregate fine (sand) in concrete mixes once.

After the polymeric blend milling process, a material with a granulometry of up to 4.75 mm was obtained and identified as fine aggregate.

The results obtained in this research show that knowing the composition and properties of individual materials, blends, and composites is essential for the definition of parameters in the processing of polymers and the use of alternative materials to obtain new products. Furthermore, the processing of a material that, as waste, accumulates in dumps, landfills, rivers, beaches, and in the marine environment, and is a problem for marine life and the population, allows adding value to waste, with the reinsertion of materials into production cycles of the polymer and composites industries.

To test the application of recycled aggregate in civil construction, the next step will be to produce concrete mixtures in which the natural aggregate (sand) is partially replaced by this material. The idea is that, among the contents evaluated, it is possible to identify the optimum content for use (the best performance) and that the mechanical and physical properties are not compromised.

## Figures and Tables

**Figure 1 polymers-15-04155-f001:**
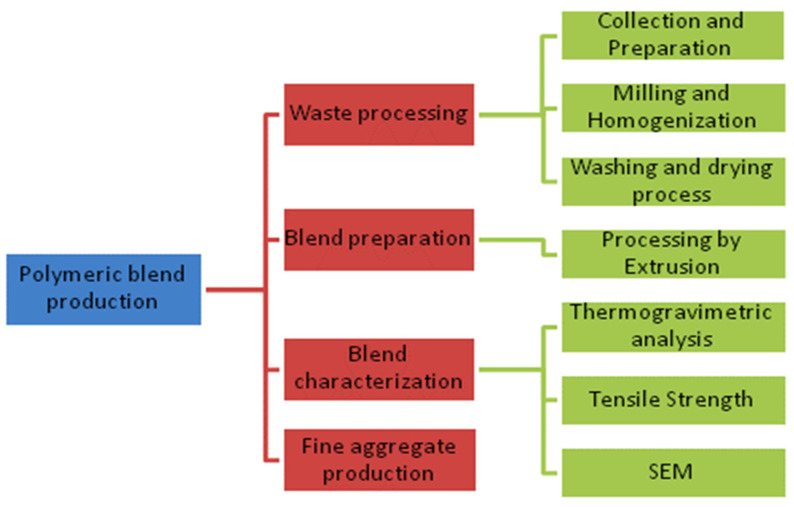
Flowchart of the methodology adopted in the recycling of plastic waste.

**Figure 2 polymers-15-04155-f002:**
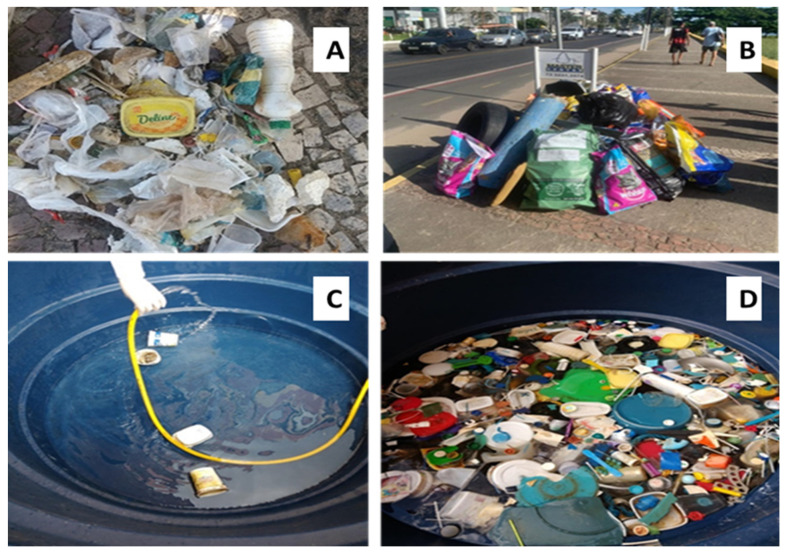
Sample plastic waste collected on beaches (**A**), separated for transport (**B**), and in the earlier washing stage (**C**,**D**).

**Figure 3 polymers-15-04155-f003:**
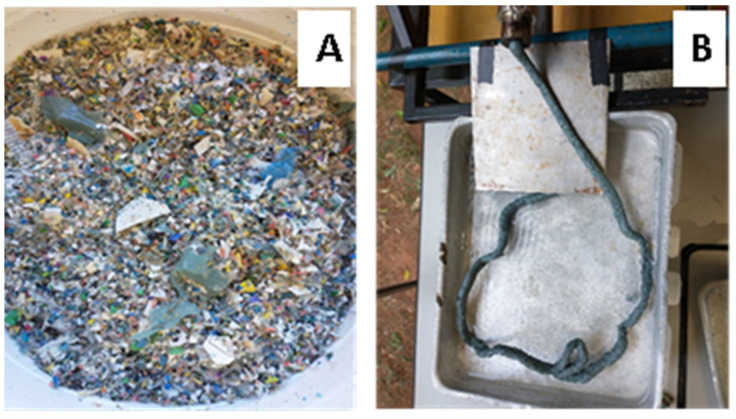
Samples of the processed plastic waste: (**A**) after milling, (**B**) after extrusion.

**Figure 4 polymers-15-04155-f004:**
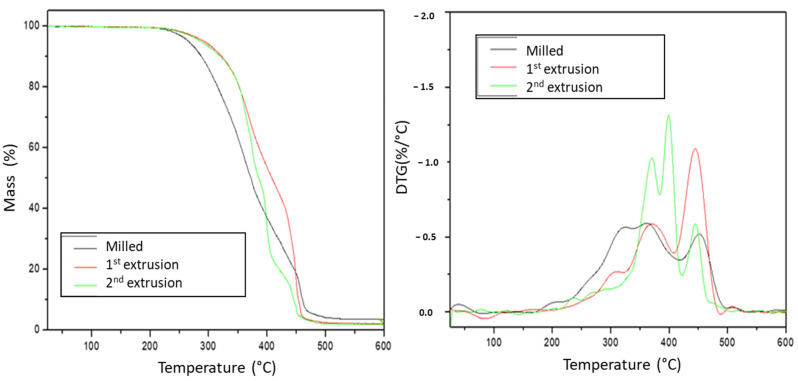
Thermogravimetry (TG) and derivative thermogravimetry (DTG) curves for polymer blends.

**Figure 5 polymers-15-04155-f005:**
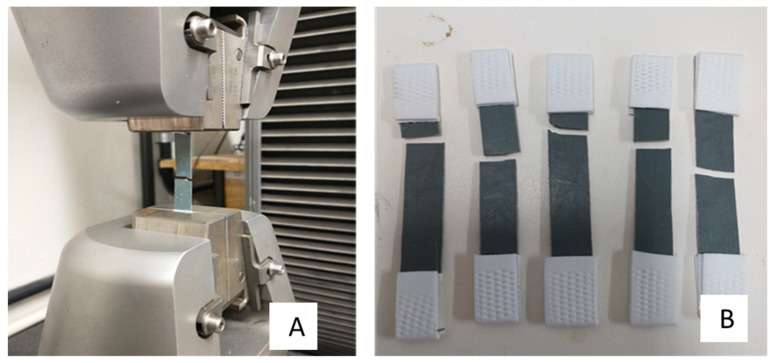
Specimens (polymer blend) prepared for tensile testing just after testing (**A**), and set of specimens showing the breaking points (**B**).

**Figure 6 polymers-15-04155-f006:**
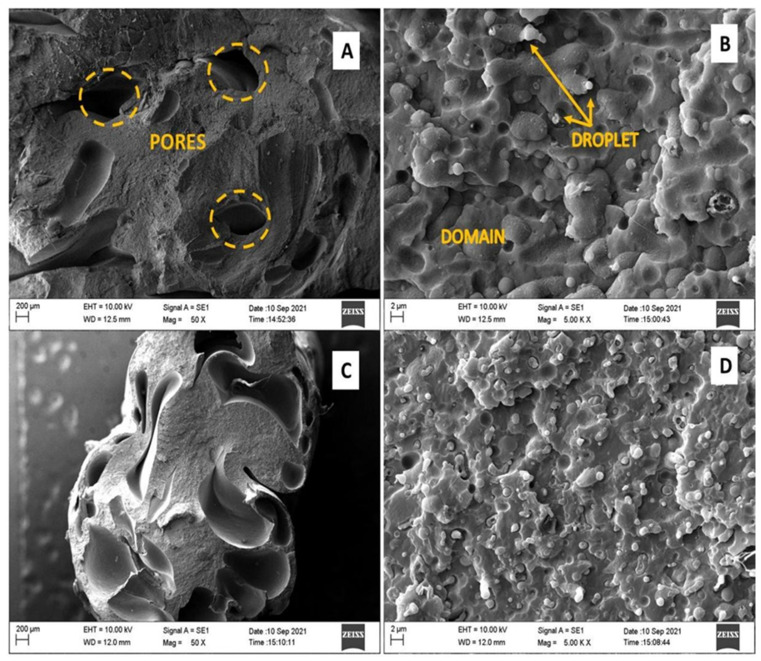
Micrographs for processed polymer blend (First Extrusion) without exit nozzle (**A**,**B**) and with exit nozzle (**C**,**D**) of the extruder. Clockwise, images were taken at 50× (**A**,**C**) and 5000× (**B**,**D**) magnifications.

**Figure 7 polymers-15-04155-f007:**
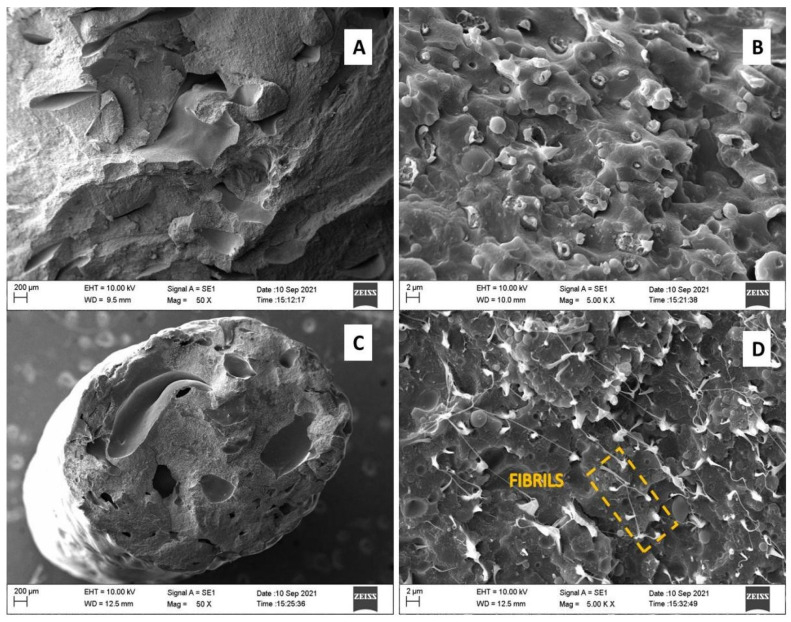
Micrographs for processed polymer blend (Second Extrusion) without the exit nozzle (**A**,**B**) and with the exit nozzle (**C**,**D**) of the extruder. Clockwise, images were taken at 50× (**A**,**C**) and 5000× (**B**,**D**) magnifications.

**Figure 8 polymers-15-04155-f008:**
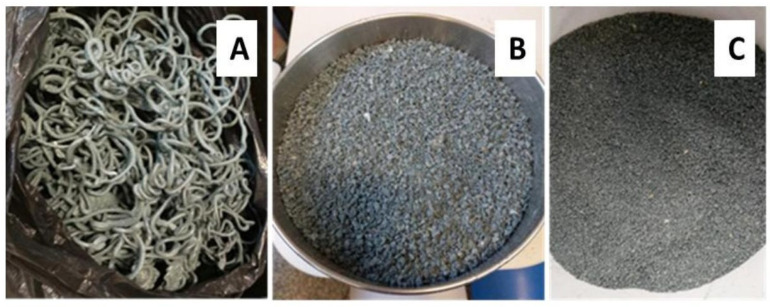
Material processed: filament after extrusion (**A**), first milling (**B**), and second milling (**C**).

**Table 1 polymers-15-04155-t001:** Presentation of plastic waste collected and amounts according to its characteristics and polymer type.

POLYMER	FEATURES	MASS (kg)
**HDPE (High-Density Polyethylene)**	Packages of cleaning and beauty products, alcohol, and bleach. Yogurt pots, buckets, bowls, and PET bottle caps. Bottles and reagents.	1.86
**LDPE (Low-Density Polyethylene)**	Plastic bags	0.30
**PP (Polypropylene)**	Plastic cups, margarine jars, plastic straws, lollipop sticks, flexible plastic rods, kitchen pots and lids, and CD packaging	2.60
**PS (Polystyrene)**	Plastic cups and straws	0.06
**OTHERS**	Toothpaste tubes, refrigerator shelves, cue cone pieces; Tupperware^®^ jars (has bisphenol A); Unidentified polymer fragments	2.02
**TOTAL**		6.84

**Table 2 polymers-15-04155-t002:** Results of the thermogravimetric analysis for the polymeric blends.

SAMPLE	T_i_ (°C)	T_f_ (°C)	T_max_ (°C)	∆T (°C) = T_max_ − T_i_	Mass (%)
**Milled residue**	239	510	362	123	96
**Blend after the first extrusion**	247	500	444	197	97
**Blend after the second extrusion**	250	490	398	148	98

**Table 3 polymers-15-04155-t003:** Properties of polymer blends obtained under different processing conditions after the tensile test.

Evaluated Properties	Processing Cycle	
	First Extrusion	Second Extrusion
Tensile strength (MPa)	10.57 ± 0.68	9.37 ± 0.20
Elastic Modulus (MPa)	1831.98 ± 227.52	2272.62 ± 427.78
Deformation (mm/mm)	1.53 ± 0.02	1.63 ± 0.11

**Table 4 polymers-15-04155-t004:** Analysis of variance1 of the effect of the amount of extrusion processing on the PP/PS/HDPE/LDPE polymer blend for the tensile strength test.

	DF	SS	SM	F	*p*
**Constant**	1	994.2968	994.2968	2965.054	0.000000
**Processing cycle**	1	3.5458	3.5458	10.574	0.011671
**Error**	8	2.6827	0.3353		
**Total**	9	6.2285			

DF = degrees of freedom; SS = squares sum; SM = squares mean; F = Fisher parameter for the significance test; *p* = significance level at 5%; Homogeneous group was determined via Duncan’s test at a 5% significance level.

**Table 5 polymers-15-04155-t005:** Results found in the literature for tensile strength, elastic modulus, and deformation for virgin PP, PS, HDPE, and LDPE, and the results obtained in this study.

Polymer	Tensile Strength (MPa)	Elastic Modulus (MPa)	Deformation at Rupture (%)
PP (Polypropylene) ^#^	31–41.4	1140–1550	100–600
PS (Polystyrene) ^#^	35.9–51.7	2280–3280	1.2–2.5
HDPE (High-Density Polyethylene) ^#^	22.1–31.0	1060–1090	10–1200
LDPE (Low-Density Polyethylene) ^#^	8.3–31.4	170–280	100–650
**Blend after first extrusion ***	10.57 ± 0.68	1831.98 ± 227.52	1.53 ± 0.02
**Blend after second extrusion ***	9.37 ± 0.2	2272.62 ± 427.78	1.63 ± 0.11

^#^ [31]; * this study.

## Data Availability

For additional information about research data, the readers can contact the correspondent author (tauanewinny@gmail.com).

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
