# Peer review of "Characterization of PS/PP/HDPE/LDPE Polymer Blend Obtained from Plastic Waste Collected on Beaches in Ilhéus-Bahia, Brazil"

_polymers, 2023, doi:10.3390/polym15204155_

Round 1
Reviewer 1 Report
The authors carried out the collection of plastic waste from beach and performed the characterization of the extrusion materials, and their processing for use as fine aggregate. Mechanical recycling of the waste plastic materials was used and the products were examined by thermal analysis, tensile testing, and SEM observation. This work showed the recycling values of polymer waste in environmental protection. Some aspects must be taken into accounts based on this work.
(1) Since the title involves the polymer blends collected from beaches, the average parameters or statistical data on many samples should be given in this work because the ingredients are varied, otherwise the title is not precise.
(2) Mechanical recycling was used for the waste treatment, the blending effect of different polymers are suspicious, the authors should introduce more details or advantages on this method.
(3) A table of the degradation parameters for each polymer may be listed for better comparison to the readers.
(4) Some testing parameters for mechanical testing might be missing, such as the stretching rate and the geometry size of the samples.
(5) The authors emphasize the homogeneous property of the material, but I’m afraid it could not be proved by SEM images only.
Some errors are found in language.
Author Response
1) Table 1 shows the materials that were collected, their composition (type of polymer and the amount present in the sample of each of the plastics.
2) In the introduction item, a paragraph about extrusion was added.
3) Table 2 presents the degradation test results for the samples, such as degradation temperature and percent mass loss.
4) Mechanical parameters have been introduced in the topic 2.5.2
Reviewer 2 Report
The study provides valuable insights into the potential of processing polymeric waste for use in civil engineering, which could have significant implications for waste management and the development of new materials. The research could benefit from further analysis of the environmental impact of the proposed approach and its long-term sustainability. Additionally, it would be helpful to explore the potential of the polymeric blend for other applications beyond its use as a fine aggregate in concrete mixes. The study presents a promising avenue for addressing the challenge of polymeric waste and underscores the importance of considering alternative materials to develop more sustainable production cycles.
1. In the abstract, it would be helpful to include a sentence or two about the significance of the research and its potential impact on the environment and/or industry.
2. In the conclusions, it would be beneficial to include a brief discussion on the potential limitations and future directions of the research. For example, what are the next steps in testing the polymeric mixture for use as fine aggregate in cementitious matrices? Are there any potential drawbacks or challenges that need to be addressed in future studies?
3. It would be helpful to include more specific details about the testing methods used to evaluate the thermal, morphological, and mechanical properties of the polymeric mixture. This information would allow readers to better understand the study design and the reliability of the results.
4. The language used in the abstract and conclusions is generally clear, but there are a few awkward phrasings that could be revised for clarity. For example, "homogeneous, rough surface, uniform in color and with the presence of voids" could be rephrased as "homogeneous with a rough surface, uniform in color, and containing voids."
Overall, this is an interesting and important study with potential applications in environmental and civil engineering.
The study provides valuable insights into the potential of processing polymeric waste for use in civil engineering, which could have significant implications for waste management and the development of new materials. The research could benefit from further analysis of the environmental impact of the proposed approach and its long-term sustainability. Additionally, it would be helpful to explore the potential of the polymeric blend for other applications beyond its use as a fine aggregate in concrete mixes. The study presents a promising avenue for addressing the challenge of polymeric waste and underscores the importance of considering alternative materials to develop more sustainable production cycles.
1. In the abstract, it would be helpful to include a sentence or two about the significance of the research and its potential impact on the environment and/or industry.
2. In the conclusions, it would be beneficial to include a brief discussion on the potential limitations and future directions of the research. For example, what are the next steps in testing the polymeric mixture for use as fine aggregate in cementitious matrices? Are there any potential drawbacks or challenges that need to be addressed in future studies?
3. It would be helpful to include more specific details about the testing methods used to evaluate the thermal, morphological, and mechanical properties of the polymeric mixture. This information would allow readers to better understand the study design and the reliability of the results.
4. The language used in the abstract and conclusions is generally clear, but there are a few awkward phrasings that could be revised for clarity. For example, "homogeneous, rough surface, uniform in color and with the presence of voids" could be rephrased as "homogeneous with a rough surface, uniform in color, and containing voids."
Overall, this is an interesting and important study with potential applications in environmental and civil engineering.
Author Response
1) The requested sentence was introduced in the summary.
2) The paragraph was introduced in the conclusion.
3) More specification of the tests performed was introduced in the methodology.
4) The terms were adjusted.
Round 2
Reviewer 1 Report
The authors have revised to the comments, even several points are not sufficient, I suggest it is deserved for publication.
The English language is OK.
Reviewer 2 Report
Accept
Accept